# INFINITETALK: AUDIO-DRIVEN VIDEO GENERATION FOR SPARSE-FRAME VIDEO DUBBING

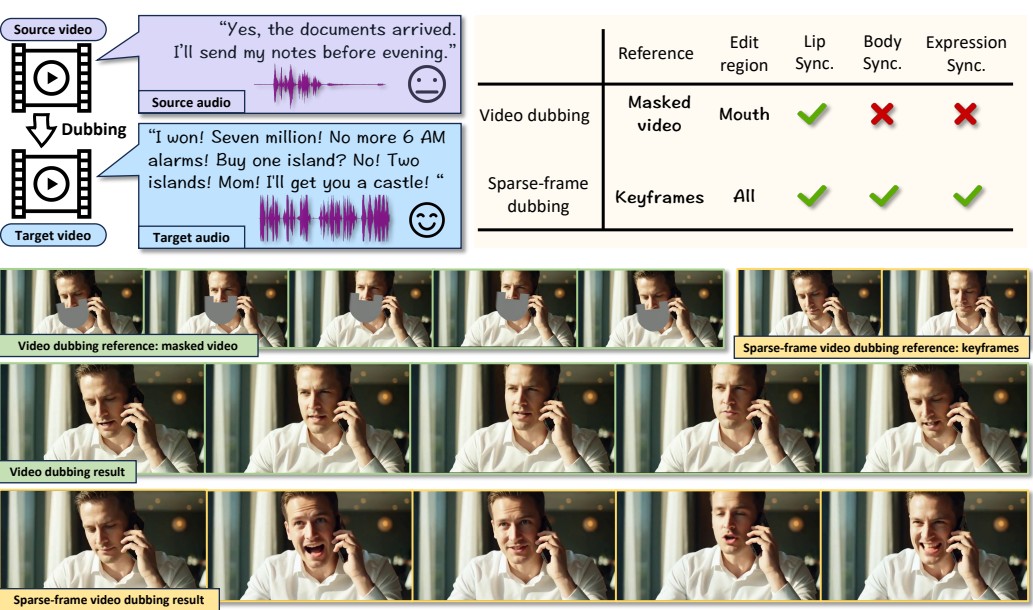

Figure 1: Compared to the traditional paradigm, sparse-frame video dubbing will not only edit mouth regions. It gives the model freedom to generate audio aligned mouth, facial, and body movements while referencing on sparse keyframes to preserve identity, emotional cadence, and iconic gestures.

## ABSTRACT

Recent breakthroughs in video AIGC have ushered in a transformative era for audio-driven human animation. However, conventional video dubbing techniques remain constrained to mouth region editing, resulting in discordant facial expressions and body gestures that compromise viewer immersion. To overcome this limitation, we introduce sparse-frame video dubbing, a novel paradigm that strategically preserves reference keyframes to maintain identity, iconic gestures, and camera trajectories while enabling holistic, audio-synchronized full-body motion editing. Through critical analysis, we identify why naive image-to-video models fail in this task, particularly their inability to achieve adaptive conditioning. Addressing this, we propose InfiniteTalk: a streaming audio-driven generator designed for infinite-length long sequence dubbing. This architecture leverages temporal context frames for seamless inter-chunk transitions and incorporates a simple yet effective sampling strategy that optimizes control strength via fine-grained reference frame positioning. Meanwhile, with additional modules, our method can also achieve acurate camera and pose control. Comprehensive evaluations on HDTF, CelebV-HQ, and EMTD datasets demonstrate state-of-the-art performance. Quantitative metrics confirm superior visual realism, emotional coherence, and full-body motion synchronization.

# 1 INTRODUCTION

Video dubbing is an audio-driven video-to-video generation task that combines an original video with new audio to create localized content Li et al. (2024); Zhang et al. (2025); Bigata et al. (2025). This process requires editing facial movements, head rotations, and body gestures to synchronize with the dubbed speech's timing and emotional tone, while preserving the source video's visual style and camera motion—capabilities essential for global media distribution.

Recent advances in audio-driven generative models have greatly improved lip synchronization for video dubbing Li et al. (2024). However, most methods focus on oral region inpainting, which causes mismatched head rotations and body gestures, reducing viewer immersion. To address this, we introduce sparse-frame video dubbing, a paradigm that preserves only key reference frames and leverages modern generative models. As illustrated in Fig. 1, our approach references select keyframes to retain the original video's emotional cadence, gestures, and camera trajectories, while allowing facial expressions, head motions, and body dynamics to organically synchronize with dubbed audio. As a long video generation task, this requires robust temporal continuation, which can be achieved by audio-conditioned video generators with initial and terminal frame guidance. Yet, naive use of audio-conditioned image-to-video generators yields unsatisfactory results, as shown in Fig. 2. These models struggle with identity preservation over extended sequences and tend to rigidly copy motion on conditioning frames, resulting in stiff facial expressions and misaligned head or lip movements. Simply conditioning on first and last frames also leads to abrupt transitions between video chunks.

To resolve these challenges, we propose InfiniteTalk, an audio-driven generator for long sequence sparse-frame video dubbing. InfiniteTalk uses a streaming video generation backbone with context frames to inject momentum and create smooth transitions. Keyframes are referenced to preserve human identity, background, and camera movement from the source video. To implement soft reference conditioning, we analyze how control strength is influenced by the similarity between video context and image condition, and propose a sampling strategy that balances control and motion alignment through fine-grained reference positioning. This enables high-quality, infinite-length dubbing with full-body, audio-aligned motion generation. Meanwhile, by using additional modules, InfiniteTalk achieves accurate camera and pose control.

We comprehensively evaluate InfiniteTalk on HDTF Zhang et al. (2021), CelebV-HQ Zhu et al. (2022), and EMTD Rang Meng (2025), covering both facial and full-body animation. Quantitative results show InfiniteTalk achieves state-of-the-art performance in audio-synchronized motion and visual quality. Human evaluation demonstrates plausible lip, face, and body movements aligned with speech cadence and emotional expression. Ablation experiments on sampling strategy and control strength further validate our algorithm's effectiveness.

Our contributions are: (1) Introducing sparse-frame video dubbing, a new paradigm for human-centric audio-driven video-to-video generation that produces natural facial expressions, head motions, and body dynamics synchronized with dubbed audio; (2) Analyzing why audio-driven image-to-video generators fall short in this task, and how reference frame positioning during training determines control strength, leading to InfiniteTalk—a streaming long-video generator with soft conditioning. We also propose the camera and pose control methods for InfiniteTalk; and (3) Extensive experiments showing state-of-the-art performance, especially in lip, head, and body motion synchronization.

# 2 RELATED WORKS

## 2.1 VIDEO GENERATION

Recent advances in generative models—including autoregressive, diffusion Wang et al. (2023b); Song et al. (2020); Nichol & Dhariwal (2021), and flow matching approaches Liu et al. (2023)—have transformed video generation. Early work such as Video Diffusion Model Ho et al. (2022b) focused on pixel-space denoising, while later methods (Make-A-Video Singer et al. (2022), PYoCo Ge et al. (2023), Imagen Video Ho et al. (2022a)) leveraged large language models for text-to-video synthesis. To address video dimensionality, research shifted to latent space learning: VideoGPT Yan et al. (2021) combined VQ-VAE Esser et al. (2021) and transformers, laying groundwork for latent modeling. Diffusion-based latent video frameworks soon emerged (He et al. (2022); Zhou et al.

(2022); Xing et al. (2023); Blattmann et al. (2023b;a); Wang et al. (2023a); Chen et al. (2023; 2024)), with CogVideoX Yang et al. (2024) introducing diffusion transformers and temporal-compressed VAEs for richer motion. Large-scale generators (Team (2024); Weijie Kong & Jie Jiang (2024); Wan et al. (2025)) now deliver unprecedented video quality.

## 2.2 AUDIO-DRIVEN HUMAN ANIMATION

Audio-driven human animation generates videos from static images, synchronizing facial and body movements to audio. Recent end-to-end diffusion-based methods Tian et al. (2024); Wei et al. (2024); Xu et al. (2024); Chen et al. (2025b); Cui et al. (2024); Ji et al. (2024); Li et al. (2024); Jiang et al. (2024) have excelled in talking head generation, eliminating intermediate representations. Other approaches extend this to full-body animation Lin et al.; Tian et al. (2025); Lin et al. (2025); Meng et al. (2024); Gan et al. (2025); Wang et al. (2025), benefiting from large, high-quality datasets. Recent works Chen et al. (2025a); Kong et al. (2025a); Huang et al. (2025) address multi-human animation. While these methods perform well on short videos, longer sequences suffer from error accumulation, such as identity loss and color drift Kong et al. (2025b). StableAvatar Tu et al. (2025) achieves infinite-length human animation but relies on a single image for conditioning, limiting its applicability for video-to-video tasks.

## 3 METHOD

### 3.1 FORMULATION

**Conditional Flow matching for audio-driven video generation**  Flow matching video generative models Liu et al. (2023); Chen et al. (2025a); Wan et al. (2025) adopts a neural network to generate realistic video frames by modeling a timestep-dependent vector field that transports samples from a noise distribution to a target video distribution. Given a ground truth conditional video distribution $q(\boldsymbol{x}|\boldsymbol{c})$ where $\boldsymbol{x} \in \mathbb{R}^{t \times h \times w \times c}$ is the encoded video latent. $\boldsymbol{c} = \{y, a, \boldsymbol{x}_{\text{ref}}\}, y \in \mathbb{R}^{m \times d_{\text{text}}}, a \in \mathbb{R}^{n \times d_{\text{audio}}}, \boldsymbol{x}_{\text{ref}} \in \mathbb{R}^{t_{\text{ref}} \times h \times w \times c}$ are the conditions, including the text prompt embedding, and the audio embedding, and the reference frames latent. Conditional flow matching defines a series of distributions by interpolating $q(\boldsymbol{x}|y, a)$ with a known trivial distribution (e.g. Gaussian noise) $p(\boldsymbol{x}|y, a)$ using a continuous variable $t \in [0.1]$.

$$q_t(\boldsymbol{x}|y, a) = (1 - t) \cdot p(\boldsymbol{x}|y, a) + t \cdot q(\boldsymbol{x}|y, a). \tag{1}$$

To be specific, a random variable $\boldsymbol{x}_t \sim q_t(\boldsymbol{x}|y, a)$ can be obtained by interpolating between $\boldsymbol{x}_0 \sim p(\boldsymbol{x}|y, a)$ and $\boldsymbol{x}_1 \sim q(\boldsymbol{x}|y, a)$ via $\boldsymbol{x}_t = (1-t) \cdot \boldsymbol{x}_1 + t \cdot \boldsymbol{x}_0$. The generative model $\boldsymbol{v}_\theta(\cdot)$, parameterized by $\theta$, is trained to match a continuous velocity field $\boldsymbol{v}_\theta(\boldsymbol{x}_t|y, a) \simeq \frac{d\boldsymbol{x}_t}{dt}$. To achieve so, we adopt the conditional flow matching objective.

$$\mathcal{L}_{\text{fm}} = \mathbb{E}_{t, \boldsymbol{x}_0, \boldsymbol{x}_1} \|\boldsymbol{v}_\theta(\boldsymbol{x}_t|y) - (\boldsymbol{x}_1 - \boldsymbol{x}_0)\|_2^2. \tag{2}$$

An ODE solver can be used to sample from a flow matching generative models.

**Sparse-frame video dubbing**  Video dubbing localizes content by replacing original audio with translated speech while preserving visual authenticity. As formalized in this work, the task transforms a source video latent $\boldsymbol{x}_0 \in \mathbb{R}^{t \times h \times w \times c}$ and a target audio $a \in \mathbb{R}^{n \times d_{\text{audio}}}$ into an output video where lip movements, facial expressions, and body dynamics synchronize organically with the new audio. Traditional video dubbing techniques focus exclusively on oral region inpainting—editing lip movements while freezing head rotations, facial expressions, and body gestures Li et al. (2024). This creates immersion-breaking mismatches, as static body language contradicts emotional speech (e.g., a rigid posture during passionate dialogue). Sparse-frame video dubbing, illustrated in Fig. 1, fundamentally redefines this process: it preserves only select keyframes $\boldsymbol{x}_{\text{ref}}$ to anchor identity, emotional cadence, symbolic gestures, and camera trajectories—critical for visual continuity—while liberating full-body dynamics (facial expressions, head motions, body gestures) to organically synchronize with dubbed audio. As Fig. 1 demonstrates, this paradigm shift enables lifelike alignment where head turns follow speech rhythm and gestures amplify emotional tone—impossible with lip-only editing. Crucially, sparse-frame dubbing operates on infinite-length sequences, demanding

Figure 2: (**left**): I2V model accumulates error for long video sequences. (**right**): A new chunk starts from frame 82. FL2V model suffers from abrupt inter-chunk transitions.

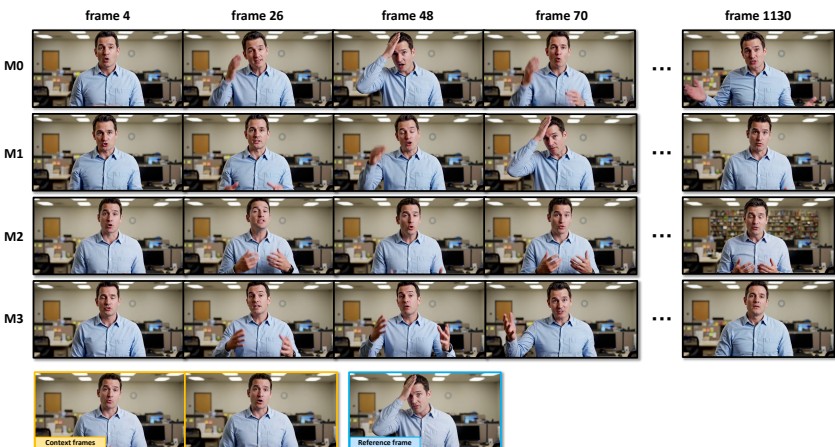

Figure 3: A visual comparison between the training reference positioning strategies. All video chunks are generated using the same context frames and the same reference frame shown in below.

generative continuation beyond short clips to maintain synchronization across extended durations, a capability unattainable with traditional frame-by-frame inpainting.

## 3.2 OBSERVATION ON NAIVE SOLUTIONS

This section explores practical strategies for sparse-frame video dubbing using two baselines: image-to-video (I2V) Cui et al. (2024) and first-last-frame-to-video (FL2V) Wan et al. (2025). As shown in Fig. 2, both face key challenges in generating long video sequences. I2V initializes each video chunk from a single reference frame and, for subsequent chunks, uses only the last frame generated. While this preserves motion flexibility, the absence of persistent anchoring to original keyframes leads to accumulated errors (e.g., gradual identity drift and shifting color tones) resulting in visible degradation. FL2V, by conditioning on both the start and end frames of each chunk, maintains alignment with source poses and eliminates accumulation errors, but enforces rigid replication of reference frames. This strict control undermines the soft conditioning needed for dubbing, where motion should adapt naturally to audio cues. Both methods also suffer from abrupt transitions between chunks due to their reliance on static frame conditions and lack of momentum continuity. Overall, I2V favors motion fluidity at the cost of accumulating errors, while FL2V ensures reference fidelity but sacrifices motion naturalness.

## 3.3 AUDIO-DRIVEN STREAMING VIDEO GENERATOR WITH REFERENCE FRAMES

To address accumulated errors in I2V models and abrupt transitions in FL2V models, we design an audio-driven streaming human animation framework. This architecture uses context frames, the trailing segment of each previously generated chunk, to propagate motion continuity via a diffusion transformer. To further prevent error accumulation, we dynamically sample multiple reference frames

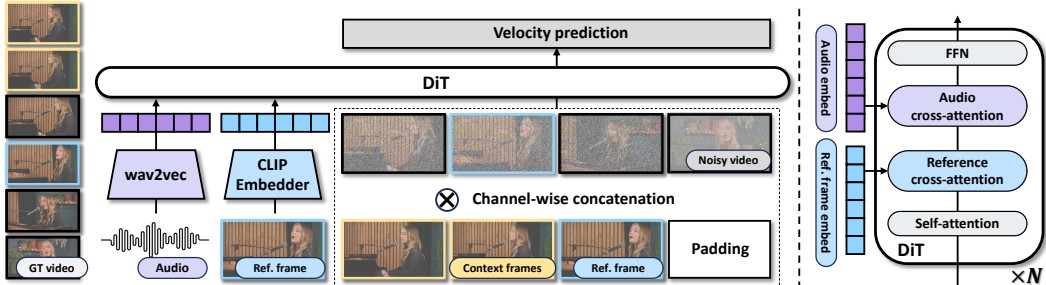

Figure 4: Visualization of InfiniteTalk pipeline. **Left**: The streaming model receives a audio, a reference frame, and context frames to denoise iteratively. **Right**: The architecture of the diffusion transformer. In addition to the traditional structures, each block includes an audio cross-attention layer and a reference cross-attention layer.

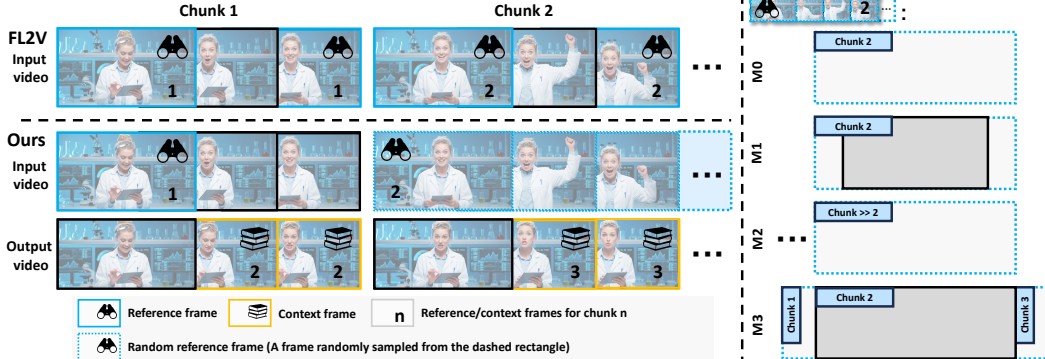

Figure 5: Visualization of reference frame conditioning strategies in video dubbing models. The top four rows show conditioning on input frames; the bottom row shows conditioning on generated frames. **Left**: Image-to-video dubbing with I2V and FL2V conditioning. **Right**: Streaming dubbing with four conditioning strategies. All approaches within each category use the same generated-video conditioning.

from the source video, preserving identity, background, camera trajectory, and style, similar to FL2V's multi-frame conditioning. Unlike FL2V's rigid frame replication, our approach supports soft conditioning as discussed in Section 3.4.

As shown in Fig. 4, the model comprises an audio embedder Schneider et al. (2019), a video VAE, and a diffusion transformer (DiT) Peebles & Xie (2022). Training requires only a video with its audio track; dubbed pairs are unnecessary. Reference frames are randomly sampled a range shown in Section 3.4, and context frames are taken as the first $4(t_c - 1) + 1$ frames. After VAE encoding, we obtain: reference frame latent $\boldsymbol{x}_{\text{ref}} \in \mathbb{R}^{c \times 1 \times h \times w}$, full video latent $\boldsymbol{x}_{\text{full}} \in \mathbb{R}^{c \times (t+t_c) \times h \times w}$, context latent $\boldsymbol{x}_{\text{context}} \in \mathbb{R}^{c \times t_c \times h \times w}$, and subsequent frames latent $\boldsymbol{x}_0 \in \mathbb{R}^{c \times t \times h \times s}$. The full video latent is $\boldsymbol{x}_{\text{full}} = \{\boldsymbol{x}_{\text{context}}, \boldsymbol{x}_0\}$, and the audio embedding $a$ is obtained from the audio sequence. For training, we use conditional flow matching at time $t$: noisy latent is $\boldsymbol{x}_t = (1 - t)\boldsymbol{x}_1 + t\boldsymbol{x}_0$ with $\boldsymbol{x}_1 \sim \mathcal{N}(\mathbf{0}, \mathbb{I})$. The DiT estimates $\boldsymbol{v}_\theta(\boldsymbol{x}_t | \boldsymbol{c})$ conditioned on $\boldsymbol{c} = \{y, a, \boldsymbol{x}_{\text{ref}}, \boldsymbol{x}_{\text{tran}}\}$. We concatenate the noisy latent and context frames in the temporal dimension ($\boldsymbol{z}_1$), pad the reference frame ($\boldsymbol{z}_2$), and concatenate these with a reference mask ($\boldsymbol{m}$) in the channel dimension:

$$\begin{aligned}
\boldsymbol{z}_1 &= \text{concat}((\boldsymbol{x}_{\text{context}}, \boldsymbol{x}_t), 2) \\
\boldsymbol{z}_2 &= \text{concat}((\boldsymbol{x}_{\text{ref}}, \mathbf{0}), 2) \\
\boldsymbol{m} &= \text{concat}((\mathbf{1}, \mathbf{0}), 2) \\
\boldsymbol{z} &= \text{concat}((\boldsymbol{z}_1, \boldsymbol{z}_2, \boldsymbol{m}), 1)
\end{aligned} \tag{3}$$

Here, $\text{concat}(\cdot)$ denotes concatenation along the specified dimension, and $\mathbf{0}, \mathbf{1}$ are zero and one tensors of appropriate size. The transformer includes cross-attention modules for both audio and

image conditioning, with reference frames processed by a CLIP vision model Cherti et al. (2023) to obtain $z_{\mathrm{ref}}$. The model is trained using the conditional flow matching loss Liu et al. (2023):

$$\mathcal{L}_{\mathrm{fm}} = \mathbb{E}_{t,\boldsymbol{x}_0,\boldsymbol{x}_1,\boldsymbol{c}}\|\boldsymbol{v}_\theta(\boldsymbol{x}_t|\boldsymbol{c}) - (\boldsymbol{x}_1 - \boldsymbol{x}_0)\|_2^2. \tag{4}$$

Next, we briefly describe the sampling method, illustrated in Fig. 5. The long video sequence is generated by auto-regressively producing small chunks. For the first chunk, the model uses the first frame of the input video as the reference, with no context frames. For subsequent chunks, the last $4(t_c - 1) + 1$ frames from the previous output serve as context frames, while the first image of the current input chunk is used as the reference frame.

### 3.4 Soft conditioning and control

**Control strength from the Reference frame**  This section investigates strategies for soft conditioning in sparse-frame video dubbing, aiming to generate audio-aligned full-body motion without rigidly mimicking reference frames that may conflict with dubbed speech. Ideally, the model should adaptively adjust control strength: applying weak control when references resemble context frames to allow motion diversity, and strong control when references differ to maintain identity and background consistency. We analyze training strategies that fulfill these requirements.

We begin with Model M0, which randomly samples reference frames from the current input chunk during training. As shown in Fig. 3, this results in excessive control, causing the model to inappropriately duplicate reference content at arbitrary times, disrupting audio-visual synchronization. To understand how reference positioning affects control, we examine chunk-level (which segment) and frame-level (which frame) sampling. Model M1 samples only the first or last frame of each chunk, mirroring FL2V's rigidity and causing the model to replicate reference poses at boundaries, even when misaligned with audio emotion. Model M2 samples from temporally distant chunks, weakening control and avoiding replication, but leading to accumulated color and background errors. Model M3 samples from adjacent chunks (within 1 second), achieving moderate control: it preserves identity and camera motion without exact duplication, and eliminates accumulated errors.

Our results show chunk-level distance is the key factor in modulating control. Short distances (M3) balance visual consistency and expressive, audio-synchronized motion; long distances (M2) destabilize outputs; fixed boundaries (M1) enforce replication and suppress dynamics. Thus, near-chunk reference sampling (M3) provides the optimal strategy for soft conditioning, enabling faithful yet flexible video dubbing.

**Camera and pose control**  We investigate camera movement preservation and pose control in sparse-frame video dubbing. Reference frames provide global control of camera trajectory, but fine-grained camera motion within video chunks may still diverge from the source. To address this, we experiment with plugins such as SDEdit Meng et al. (2022) and Uni3C Cao et al. (2025), but find they alone cannot maintain scene information during large camera movements, as a single reference frame is insufficient for full scene coverage. Therefore, we extend InfiniteTalk to use two reference frames when cloning camera trajectory and background is needed. Additionally, we finetune a UniAnimate Wang et al. (2024) module for pose control within InfiniteTalk. Further details are provided in the appendix.

## 4 Experiment

**Implementation details**  Our model builds on MeiGen-MultiTalk Kong et al. (2025a), featuring a 14B-parameter DiT for audio-driven image-to-video generation at multiple resolutions. We use wav2vec2 Baevski et al. (2020) for audio embedding and CLIP/H Cherti et al. (2023) for image embedding. Following OmniHuman Lin et al. (2025), we collect 2,000 hours of talking person videos from the internet. Training is performed on a cluster of 64 NVIDIA H100 80G GPUs. Each context includes 9 images, giving $t_c = 3$ context frames in latent space. Video chunks are 81 frames long, and the model generates 72 frames per autoregressive step for long videos.

**Test datasets and evaluation metrics**  We evaluate our method on HDTF Zhang et al. (2021), CelebV-HQ Zhu et al. (2022) (facial dynamics), and EMTD Rang Meng (2025) (full-body movement).

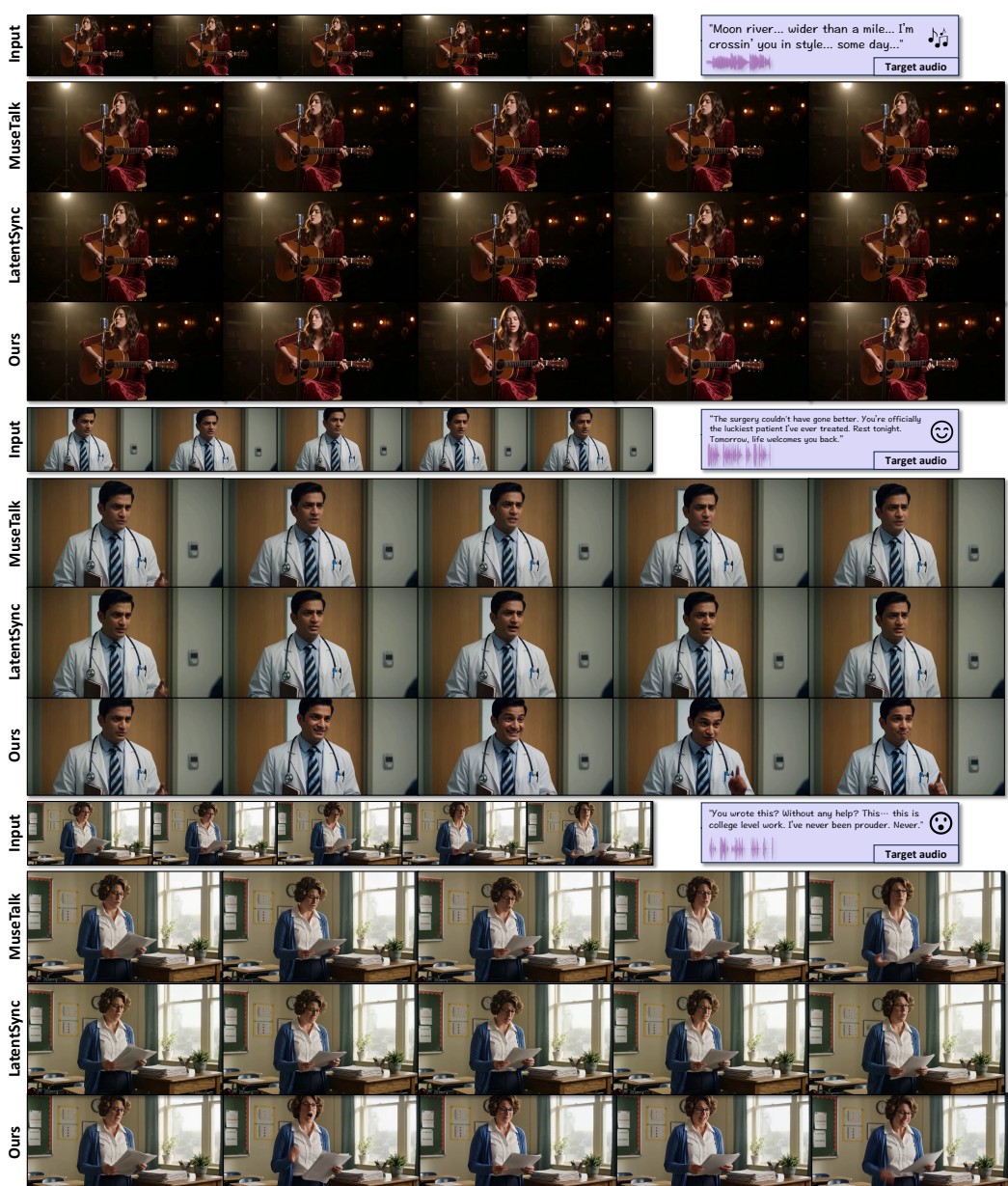

Figure 6: A visual comparison between the video dubbing methods.

Following dubbing protocols Li et al. (2024); Fei et al. (2025), we create a test set of 120 videos by sampling 40 per dataset and permuting audio channels to simulate dubbing. Videos are dubbed at $480 \times 480$ resolution, matching the input frame count. Performance is assessed using both automatic and human metrics. Objective metrics include FID (visual quality), FVD (temporal coherence), SyncNet's Sync-C and Sync-D (lip sync), and CSIM (identity preservation). Human studies cover gesture-audio sync, head-speech alignment, lip sync, identity consistency, and overall naturalness, with 340 responses from 17 participants on all 40 EMTD dubbing results.

## 4.1 QUANTITATIVE EXPERIMENTS

We compare InfiniteTalk to traditional video dubbing methods (MuseTalk Zhang et al. (2025), FantacyTalking Wang et al. (2025), Hallo3 Cui et al. (2024)) and audio-driven image-to-video models

| Dataset | Model | Metrics | | | | |
|---|---|---|---|---|---|---|
| | | FID↓ | FVD↓ | Sync-C↑ | Sync-D↓ | CSIM↑ |
| HDTF | LatentSync | 16.09 | **48.45** | 8.97 | 7.13 | 0.916 |
| | MuseTalk | **14.20** | 49.13 | 7.17 | 7.90 | **0.933** |
| | Ours | 26.11 | 131.65 | **9.35** | **6.67** | 0.775 |
| CelebV-HQ | LatentSync | 17.80 | **67.97** | 6.90 | 7.47 | **0.869** |
| | MuseTalk | **17.62** | 72.07 | 4.16 | 9.86 | 0.857 |
| | Ours | 32.29 | 229.67 | **7.53** | **7.33** | 0.726 |
| EMTD | LatentSync | **11.43** | 212.60 | 8.13 | 7.29 | **0.846** |
| | MuseTalk | 14.26 | **46.07** | 5.35 | 9.28 | 0.825 |
| | Ours | 32.55 | 312.17 | **8.60** | **7.16** | 0.713 |

Table 1: Quantitative comparisons our methods between the traditional video dubbing models.

| Dataset | Model | Metrics | | | | |
|---|---|---|---|---|---|---|
| | | FID↓ | FVD↓ | Sync-C↑ | Sync-D↓ | CSIM↑ |
| HDTF | FantacyTalking | 32.06 | **110.36** | 3.78 | 10.80 | 0.684 |
| | Hallo3 | 36.48 | 144.65 | 7.20 | 8.61 | 0.674 |
| | OmniAvatar | 26.63 | 112.49 | 7.06 | 8.63 | 0.752 |
| | MultiTalk | 27.61 | 133.58 | 9.02 | 6.96 | **0.754** |
| | StableAvatar | 29.21 | 141.63 | 8.56 | 7.86 | 0.751 |
| | Ours | **27.14** | 132.54 | **9.18** | **6.84** | 0.751 |
| CelebV-HQ | FantacyTalking | 37.53 | 237.58 | 2.93 | 10.79 | 0.654 |
| | Hallo3 | 42.36 | 258.65 | 5.63 | 9.12 | 0.591 |
| | OmniAvatar | 37.41 | 250.67 | 5.88 | 8.68 | 0.703 |
| | MultiTalk | 34.79 | 230.41 | 7.25 | 7.70 | 0.711 |
| | StableAvatar | 34.23 | 235.79 | 7.06 | 8.01 | 0.709 |
| | Ours | **33.96** | **230.12** | **7.41** | **7.59** | **0.713** |
| EMTD | FantacyTalking | 36.66 | **298.24** | 3.60 | 11.31 | 0.626 |
| | Hallo3 | 44.71 | 326.94 | 5.68 | 9.56 | 0.512 |
| | OmniAvatar | **29.47** | 308.14 | 6.93 | 8.55 | 0.694 |
| | MultiTalk | 33.80 | 315.33 | 8.13 | 7.50 | 0.702 |
| | StableAvatar | 34.64 | 331.40 | 7.84 | 7.69 | 0.699 |
| | Ours | 33.27 | 314.68 | **8.34** | **7.36** | **0.709** |

Table 2: Quantitative comparisons between our method and audio-driven image-to-video models.

(OmniAvatar Gan et al. (2025), MultiTalk Kong et al. (2025a), StableAvatar Tu et al. (2025)), using their open-source weights and inference scripts for consistency.

As shown in Table 2, InfiniteTalk significantly outperforms image-to-video models in lip synchronization. However, there is a trade-off among synchronization (Sync-C, Sync-D), visual quality (FID, FVD), and identity preservation (CSIM). Methods that simply copy the input achieve top scores in FID, FVD, and CSIM, but do not reflect true visual quality. When compared to methods with competitive synchronization, InfiniteTalk excels in both visual quality and identity preservation. In Table 1, traditional dubbing methods like LatentSync Li et al. (2024) and MuseTalk Zhang et al. (2025) only edit the oral region, leaving the rest of the video unchanged and thus inflating FID and FVD scores. Currently, no automatic metric effectively measures full-body motion and audio alignment; music-motion metrics and Sync-C/Sync-D fail when there is significant head movement. To address this, we conduct a user study, as shown in Table 4, where participants rank dubbing results from MuseTalk, LatentSync, and our method. InfiniteTalk achieves the highest scores in both lip and body motion synchronization, highlighting the limitations of traditional methods that restrict edits to the mouth and often misalign body motion.

| Model | FID↓ | FVD ↓ | Sync-C↑ | Sync-D↓ |
|---|---|---|---|---|
| Ours (M0) | 32.69 | 322.04 | 8.51 | 7.31 |
| Ours (M1) | **32.21** | **307.21** | 7.96 | 8.11 |
| Ours (M2) | 42.17 | 376.53 | 8.23 | 7.44 |
| Ours (M3) | 32.55 | 312.17 | **8.60** | **7.16** |

Table 3: Ablation experiment results on EMTD.

| Model | Lip Sync.↓ | Body Sync. ↓ |
|---|---|---|
| MuseTalk | 2.57 | - |
| LatentSync | 2.32 | 1.92 |
| Ours | **1.11** | **1.09** |

Table 4: Human evaluation between video dubbing methods on motion synchronization. Body sync for MuseTalk is omitted as it is identical to LatentSync.

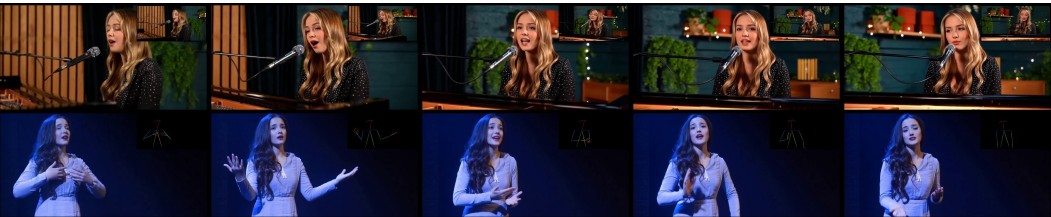

Figure 7: InfiniteTalk can achieve camera and pose control. The upper-right corners show the condition.

## 4.2 QUALITATIVE EXPERIMENTS

We conduct a visual comparison between our method and traditional video dubbing methods in Fig. 6. The first input example is a static video. It showcases when only editing mouth regions, traditional video dubbing methods cannot drive the head and body by the audio track. The following two inputs are dynamic videos. Compared to the counterparts, InfiniteTalk is not only able to generate plausible audio-aligned lip movements, but also synchronized face, head, and body movements with matched emotional expressions. As a new paradigm, sparse-frame video dubbing also demonstrates its necessity for modern audio-driven human animation video-to-video applications. The visual results showing the camera and pose control capability is shown in Fig. 7. The fine-grained level of customization ensures reliable controllable sparse-frame video dubbing.

## 4.3 ABLATION STUDY

To systematically evaluate which training strategy performs the best in sparse-frame video dubbing, we conduct ablation studies comparing the four different reference frame positioning methods introduced in Section 3.4. All ablated models are rigorously benchmarked using the 40-video test set sampled from EMTD under identical conditions. As shown in Table 3, our fine-grained reference frame positioning during training is the key to achieve reliable visual quality and audio-motion synchronization.

## 5 CONCLUSION

We introduce sparse-frame video dubbing, a novel paradigm for audio-driven video-to-video generation that employs reference keyframes to maintain emotional cadence and camera trajectories while liberating facial, head, and body dynamics to synchronize organically with dubbed audio. We propose InfiniteTalk, an audio-driven generator that overcomes critical limitations in long-form synthesis. By incorporating transient frame conditioning for seamless transitions, motion-provoking sampling to activate natural gestures, and adaptive camera control, InfiniteTalk achieves state-of-the-art lip, head, and body synchronization while eliminating identity drift and motion artifacts across extended sequences. Extensive validation confirms its superiority in producing natural, audio-aligned dynamics essential for immersive dubbed content.

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

## A    LLM USAGE

DeepSeek R1 and GPT 4.1 are used to only polish the text on the finished manuscript. LLama-Next is used to generate textual prompts of the videos.

## B    CONTROLLED SPARSE-FRAME VIDEO DUBBING

**Soft first-last frame conditioning**    To improve the content preserving in the scene, we extend InfiniteTalk, enabling it to use two reference frames to confirm the details in the video chunk. The model is finetuned from the M3 model. The two reference images are randomly selected from the M3 reference range shown in Fig. 5, then we sort the two frames according to their frame number. Let $x_{ref1}$ and $x_{ref2}$ denote the first reference frame and the second reference frame respectively, the conditional input of the InfiniteTalk extension is:

$$\begin{aligned}
z_1 &= \mathrm{concat}((x_{context}, x_t), 2) \\
z_2 &= \mathrm{concat}((x_{ref1}, \mathbf{0}, x_{ref2}), 2) \\
m &= \mathrm{concat}((\mathbf{1}, \mathbf{0}), 2) \\
z &= \mathrm{concat}((z_1, z_2, m), 1).
\end{aligned} \tag{5}$$

We do not feed the second reference frame to the clip vision embedding. The model is finetuned by only updating audio cross-attention layers. The M3 reference range ensures soft conditioning of the two reference frames. We can achieve better scene consistency without losing introducing rigid replication on the content of reference frames.

**Camera trajectory control**    We test our model with SDEdit Meng et al. (2022) and Uni3C Cao et al. (2025). SDEdit incorporates trajectory information by adding the source video to the initialize noise at a scale $t_0$. The denoising sampling process starts from $t = t_0$ instead of $t = 1$. $x_{t_0} = (1 - t_0) \cdot x_1 + t_0 \cdot x_0$. While Uni3C is a ControlNet-like architecture attached to the diffusion transformer. A visual comparison is shown in Fig. 8. Using InfiniteTalk alone will not replicate the subtle camera movement of source video. With SDEdit Meng et al. (2022) or Uni3C Cao et al. (2025), we can achieve fine-grained camera control.

**Pose control**    To enable InfiniteTalk for pose-controlled human animation. We borrow the modules from VACE Jiang et al. (2025) and UniAnimate Wang et al. (2024). Compared to VACE, we find UniAnimate is introducing very little additional computation, since it only perform low-rank adaptation on the diffusion transformer, whereas VACE rely on a ControlNet architecture. To further enhance the pose control in our audio-driven animation task. We finetune the model using LoRA on 1,000 videos given the audio and pose annotation.

## C    ADDITIONAL RESULTS

**Pose control**    We show examples using human pose to drive sparse-frame video dubbing in Fig. 9. InfiniteTalk allows users to use a custom human motion sequence to control the generated video. Our method will generate satifactory facial and body details to match the pose input and the audio, allowing versatile human related content creation.

**Long video dubbing**    We show visual results for long video generation using InfiniteTalk. In Fig. 10 and Fig. 11, we use a single image as the condition and use a long audio track to animate the human in the condition image. Our method is able to achieve high fidelity full body animation. After thousands of frames, InfiniteTalk will not cause color tone or identity bias. In Fig. 12 and Fig. 13, we dub a long video using a long video track. Our method can follow the camera movements, scene cut of the input video while produce authentic, audio-aligned human motion. Please see our attachment for additional video samples.

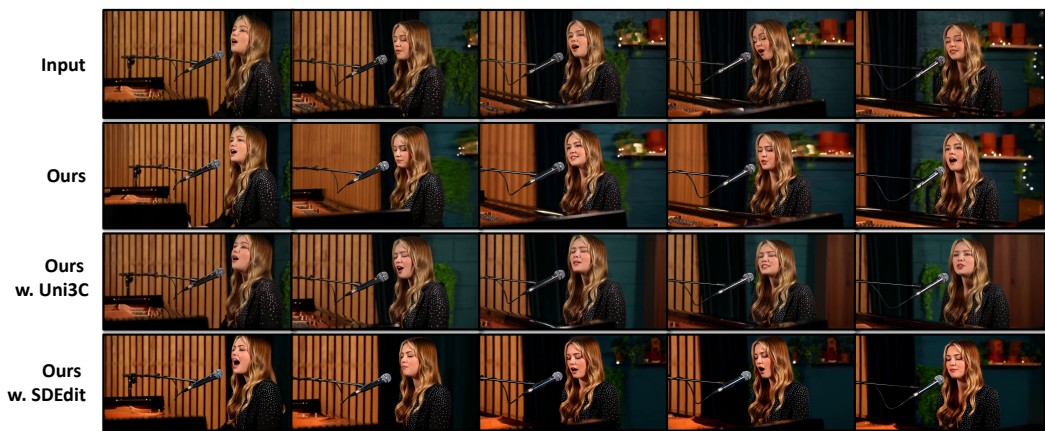

Figure 8: Visual comparison on the camera trajectory control methods.

Figure 9: Pose controlled sparse-frame video dubbing.

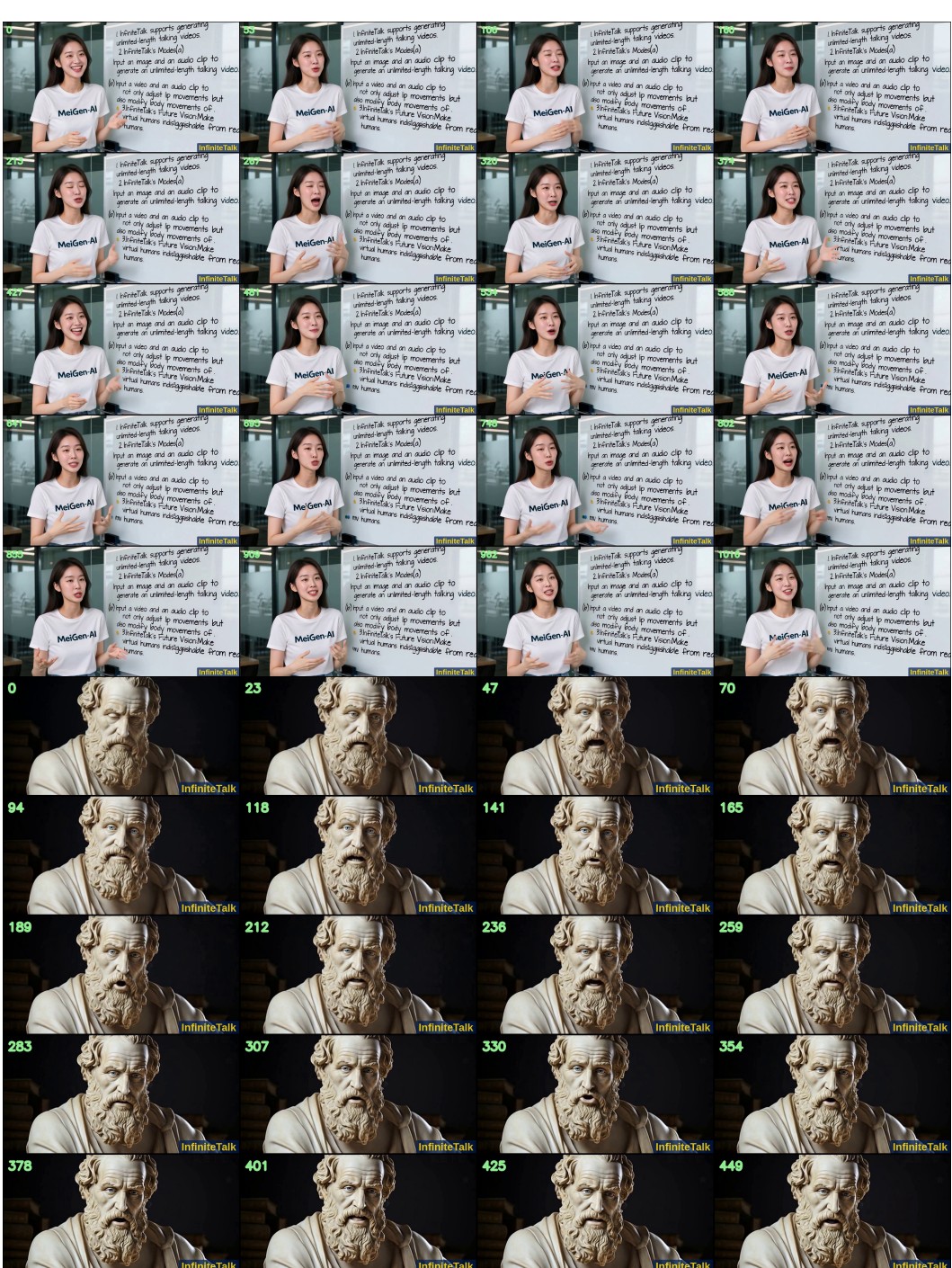

Figure 10: Long image animation results. The number in left up corners show the frame number.

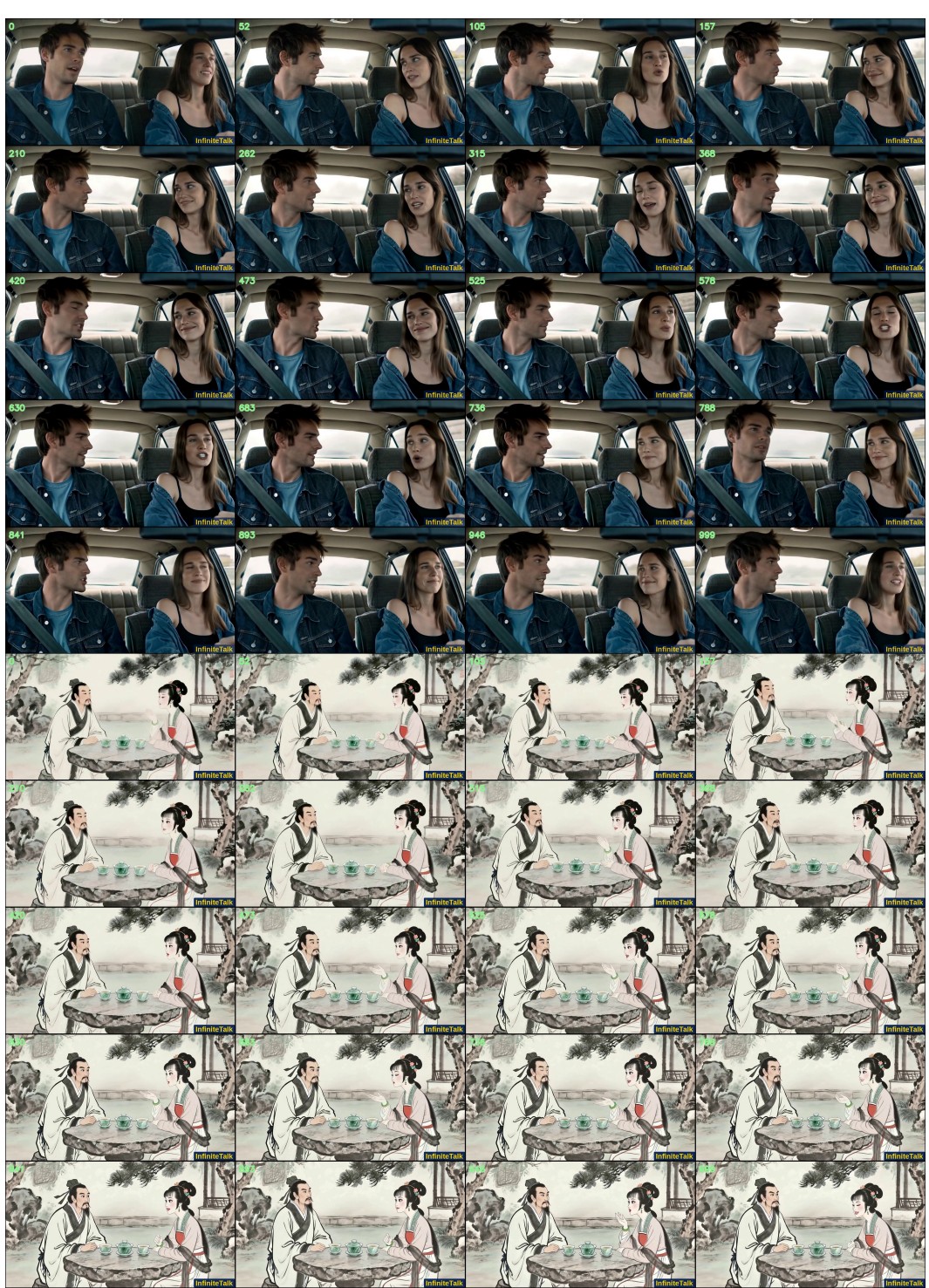

Figure 11: Long image animation results. The number in left up corners show the frame number.

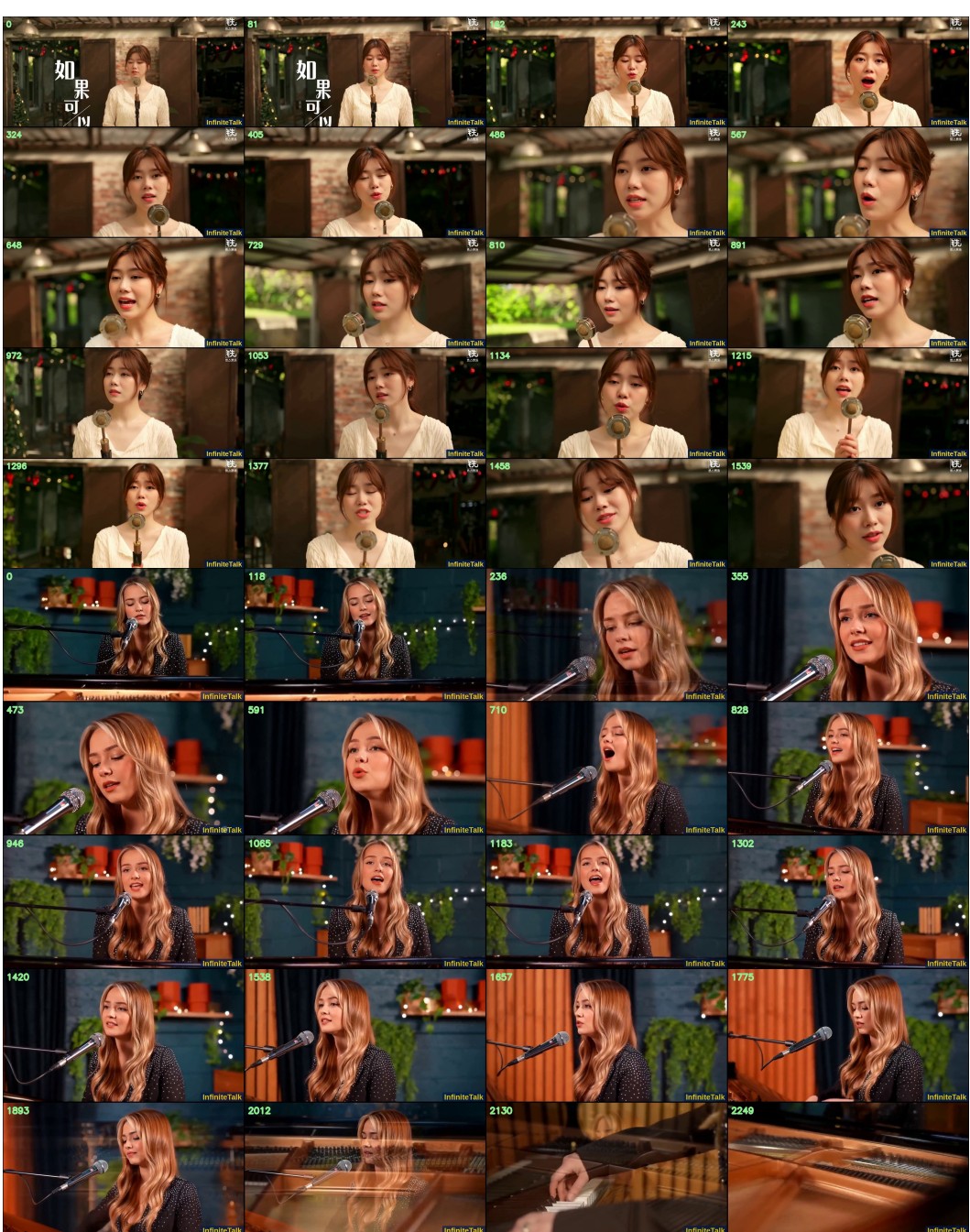

Figure 12: Long video dubbing results. The number in left up corners show the frame number.

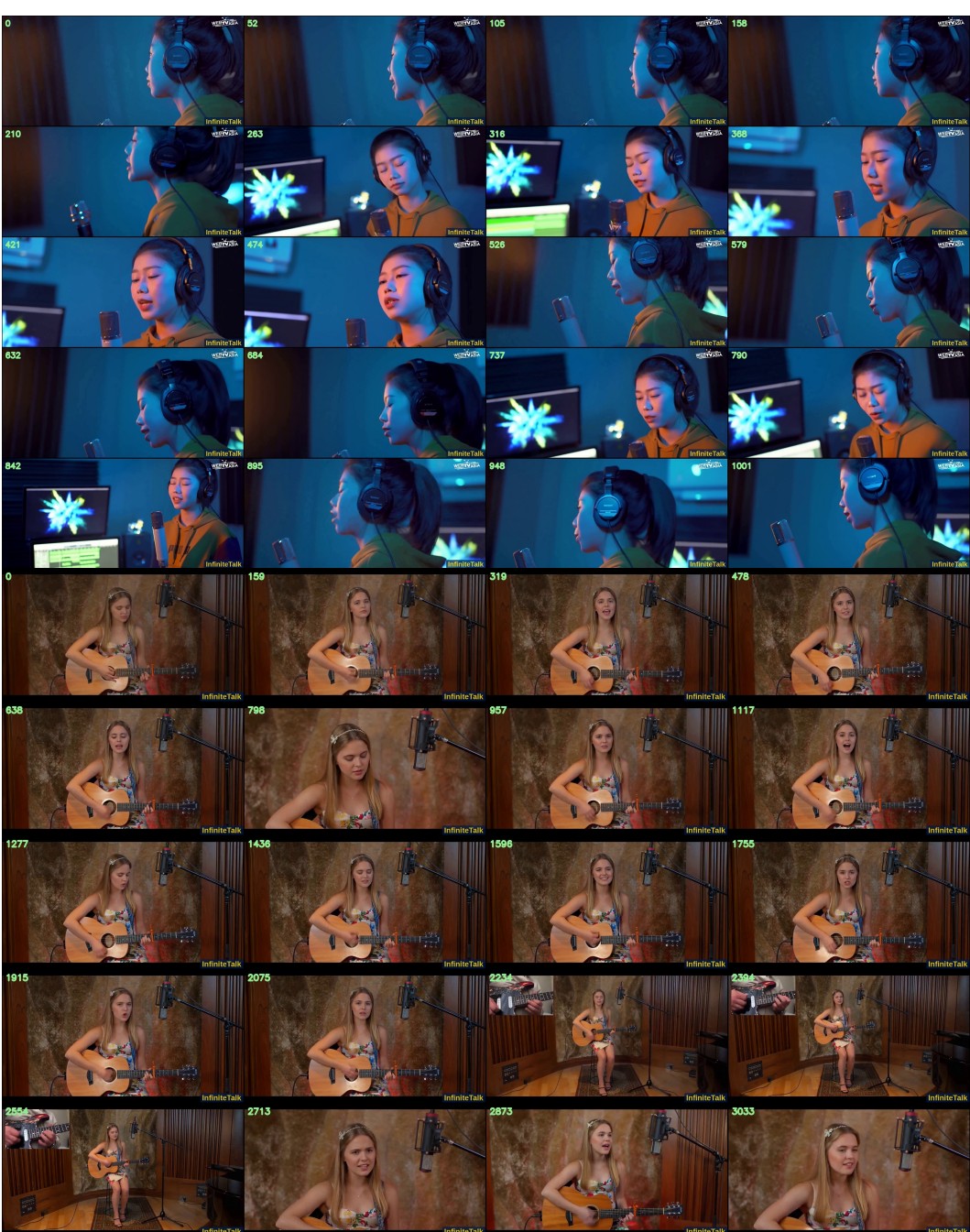

Figure 13: Long video dubbing results. The number in left up corners show the frame number.

