# OpenReview forum: "InfiniteTalk: Audio-driven Video Generation for Sparse-Frame Video Dubbing"
_ICLR.cc/2026/Conference — ICLR 2026 Conference Withdrawn Submission_

### Official Review · Reviewer_A7jy · 2025-10-21

**Soundness:** 2
**Presentation:** 3
**Contribution:** 2
**Rating:** 4
**Confidence:** 4

**Summary:**

This paper finds that the I2V and FL2V models both cannot address long video generation well. There would be color difference and identity difference accumulation in auto-regressive generation process. And the FL2V will lead to hard chunk switching and inconsistent motion. So this paper analyzes the impact of reference frame positioning on control intensity, and proposes a strategy of sampling reference frames from adjacent blocks. This strategy achieves a balance between "weak control" (ensuring motion diversity when reference frames are similar to context frames) and "strong control" (maintaining identity and background consistency when reference frames differ significantly).

**Strengths:**

1.	The exploration of the reference strategy is relatively detailed, which is inspiring for other works in the field.
2.	The chunk transition in the demo video is relatively natural, and traces of chunk switching are hardly noticeable. Moreover, with a duration of approximately one minute, the video does not show obvious error accumulation.
3.	The quantitative experiments have achieved results comparable to SOTAs.
4.	The writing and presentation are clear and easy to understand.

**Weaknesses:**

1.	The proposed task seems to be confusing. The proposed sparse video dubbing task requires the original video to obtain sparse key frames, and the generated length depends on the length of the original video. However, the contribution claimed in the paper is infinite-length video generation, which conflicts with the task setting since there are no “infinite” images in a certain video. Additionally, the method in the paper does not include content related to key frame planning and generation.
2.	It seems that no quantitative comparison has been conducted with other FL2V methods. From my perspective, in terms of the content and type of input and output, the task closest to the sparse video dubbing task is FL2V rather than traditional video dubbing. The paper claims that FL2V methods have the problem of unnatural motion transitions, which requires sufficient evidence to prove, and merely presenting Figure 2 is insufficient.
3.	The quality of some cases in the supplementary materials is not very satisfactory. For example, pose-driven video-0 has multiple and obvious hand distortions.
4.	The paper lacks a quantitative evaluation of "abrupt transitions", which is claimed to be bad when using FL2V in Lines 72-73. As an important part of the paper, a direct quantitative method should be designed to evaluate this indicator.

**Questions:**

1.	Explain the novelty of the proposed task.
2.	Conduct comparisons with FL2V methods.
3.	Explain why some demos have bad video quality.
4.	Add direct quantitative metrics about "abrupt transitions".

---

### Official Review · Reviewer_mhFe · 2025-10-24

**Soundness:** 4
**Presentation:** 3
**Contribution:** 3
**Rating:** 4
**Confidence:** 4

**Summary:**

This paper presents sparse-frame video dubbing (a new audio-driven video-to-video paradigm) and its streaming generator InfiniteTalk, tackling traditional dubbing’s flaw of only editing mouth regions (causing mismatched motions and poor immersion). Unlike rigid baselines (e.g., I2V, FL2V) with error accumulation or abrupt transitions, it retains source keyframes for identity/camera consistency while enabling full-body audio alignment.

Key contributions: Adapt "context frame" strategy to dubbing; A soft conditioning strategy balancing source consistency and motion flexibility; Dedicated modules for precise camera trajectory/pose control.

Experiments on HDTF, CelebV-HQ, EMTD show InfiniteTalk achieves SOTA lip/body synchronization and identity preservation, outperforming MuseTalk, LatentSync in metrics and human evaluations.

**Strengths:**

1. InfiniteTalk proposes the "sparse-frame video dubbing" paradigm. It breaks traditional "mouth-only edit" limitations, enabling full-body audio-aligned motion while preserving the source video’s identity, scene, and camera trajectory.
2. The paper adapts context frame and reference frame strategies to dubbing needs. Context frames focus on dynamic motion, bind to audio via cross-attention, and solve long-sequence abruptness and motion-audio mismatch.
3. Its experiments are rigorous, using 3 datasets (HDTF, CelebV-HQ, EMTD) for facial/full-body animation. It compares with over 10 baselines and uses ablation studies to validate key modules like M3 sampling.
4. InfiniteTalk enables long-sequence video dubbing via its streaming autoregressive framework, addressing the long-standing issue of error accumulation
5. The paper is well-written and clearly presents its objectives, methodology, and findings.

**Weaknesses:**

1. While applied to video dubbing, the core reference-based long-sequence generation strategy is already common in I2V, limiting the work’s originality.
2. Supplementary results omit camera control examples, and the analysis of limitations (e.g., motion repetition, efficiency) is insufficient.
3. It would be good to visualize the ablation study for the soft reference conditioning.

**Questions:**

1. "6.mp4" in the "long image animation" section of the supplementary materials still shows error accumulation. Does this model have performance biases in domains other than real humans?
2. What is the role of the reference mask mentioned in the paper? Is it aligned with the reference image at the frame-level?
3. Taking long video generation from a single image as an example, color issues still occur even with a reference image. The paper mentions that modifying the soft reference conditioning from M2 to M3 can alleviate error accumulation in long-term continuation. Does this mean that other single-image methods can also adopt similar modifications to mitigate continuation issues? Or are there other modification directions or observational findings that affect continuation performance?
4. What are the known limitations or specific scenarios where InfiniteTalk struggles? Highlighting these would give a more complete picture of the model’s strengths and areas for improvement.

---

### Official Review · Reviewer_wfLr · 2025-10-26

**Soundness:** 3
**Presentation:** 3
**Contribution:** 3
**Rating:** 6
**Confidence:** 4

**Summary:**

The paper introduces InfiniteTalk, a framework under the newly defined paradigm called Sparse-frame Video Dubbing (SFVD).
Unlike traditional dubbing methods that modify only the mouth region, InfiniteTalk generates full-body motion synchronized with dubbed audio, while preserving identity, gestures, and camera trajectory using sparse reference frames.
The method is based on a flow-matching diffusion transformer with context frames for temporal continuity and a soft conditioning strategy that modulates control strength according to reference-frame distance.

**Strengths:**

1. Novel Problem Formulation: the notion of “sparse-frame video dubbing” is interesting and it generalizes mouth-region editing to holistic, full-body motion editing in dubbing tasks.

2. Additional modules (SDEdit, Uni3C, UniAnimate) show good extensibility for camera and pose control.

**Weaknesses:**

1. Unclear Methodological Details: equation (3) lacks explicit tensor dimension consistency; although fig 5 looks fancy, actually it doesn't help readers to understand the different type of reference position strategy.

2. Lack enough experiments: how the frames of context affect the continuity; the effectiveness comparison between the reference cross-attention and channel-wise concatenation.

3.  Overstatement of Generalization and “Infinite-Length” Claims: “Infinite-length” generation is actually chunk-wise autoregressive v2v synthesis—not continuous streaming as implied.

4. Missing Discussion of Failure Modes: No section or analysis discusses limitations (e.g., when the model drifts, fails to preserve context, or how errors manifest over very long videos).

5. Trade-Offs in Quantitative Results: The paper’s core quantitative comparisons (Tables 1 and 2) reveal that InfiniteTalk, while strong in motion synchronization, falls behind baselines on certain key metrics: FID and FVD scores are worse than mouth-region dubbing methods, and identity preservation (CSIM) is sometimes lower. For instance, Table 1 shows a clear drop in CSIM versus LatentSync/MuseTalk, and in Table 2, improvements over strong image-to-video baselines are modest. The argument that ‘methods that simply copy the input achieve top scores in FID, FVD, and CSIM, but do not reflect true visual quality’ (Section 4.1) is only partially convincing; better discussion or alternative metrics targeting full-body, full-face realism are warranted. This trade-off is critical, as practitioners may still prioritize photorealism and identity conservation.

6. Camera/Pose Control operation: Extensions for trajectory/pose control (Section 3.4, and Appendix B) are alluded to, but lack a full mathematical or implementation description.

**Questions:**

1. The meaning of x_tran in Line 261
2. Why doesn't the condition in Equation(1) has x_ref?

---

### Official Review · Reviewer_vhtA · 2025-10-31

**Soundness:** 3
**Presentation:** 2
**Contribution:** 3
**Rating:** 4
**Confidence:** 4

**Summary:**

This paper claims a better video dubbing model by replacing inputs from the the whole masked video to several keyframes (images), and proposing a long video generation method by using multiple reference frames, e.g. 9 reference frames.

The base model / backbone is Wan2.1 14B and pretrained on MultiTalk's in house 2k hours dataset. The evaluation is on open-source dataset such as HDTF.  The performance are SoTA for both objective and subjective metrics.

The main contribution is the performance and open-source for the model weights.
The main weakness is it is a long video generation paper for ATI2V model. but the proposal (using multiple reference frames) is not new and needs more comparsions for the same propose methods.

**Strengths:**

1. the objective and subjective results quality are good,
2. the models are open-source, which is useful for the community.
3. the paper is organized and could follow.
4. the video results from supp. are sufficient.

**Weaknesses:**

1. Better Video Dubbing or Audio Contioned Text-Image to Video (ATI2V) model?  the motivation of this paper is propose a better video dubbing that could edit more than lip region. But from my understanding video dubbing and ati2v are different tasks, you may only want to edit the lip region and keep the body gesture from original actor, for example, video dubbing or post editing for films. I think it is unfair to compare with video dubbing models. Besides, in general any ati2v models have this advantage that could edit body, this may not from the proposal in this paper. overall I may need more reasons to agree the contribution (i) in paper.
2. Using multiple reference frames for long video is not new and lack of comparison.
    - the main problem the authors want to address is error accumulation in long video. But the comparions are only with single chunk baselines. there are several papers discussed this topic in general video generation, e.g., framepack. Also in ATI2V model, e.g. magicinfinite.
    - using multiple frames instead of single frames is a strightforward baseline. for example, magicinfinite also proposes use 33 frames overlap window (reference frames). The differneces between M0-M3 are also intuitive - instead of using reference frames continously, using them by skip one chunk.
    - lack of ablation of key parameters such as number of reference frames etc. for example, at least how many frames could ensure the velocity / acceleration consistency? the reconstructed reference frame maybe not as same as the input, is there any blending here?
3. Reproducibility. The main contribution of the performance may from the filtered training data. But there is a lack of data processing or key filitering parameters. Making the reproducibility is limited.

**Questions:**

Thanks to the author releasing their weights to the community, overall this is a valuable work but my major concern is this is a long video generation paper, and the proposal cant convince me it is better than concurrent works. I am glad to raise my score if author correct my understanding on this, both in theroy or on results are fine.

---

### Official Review · Reviewer_gKYA · 2025-11-01

**Soundness:** 2
**Presentation:** 2
**Contribution:** 2
**Rating:** 2
**Confidence:** 4

**Summary:**

This paper presents an approach to audio-driven video generation, specifically for sparse-frame video dubbing. This method aims to address the limitations of traditional video dubbing techniques, which primarily focus on lip-syncing while neglecting the synchronization of  head movements and body gestures with the dubbed audio. The proposed approach leverages sparse-frame video dubbing, where key reference frames are preserved to maintain visual consistency, identity, gestures, and camera trajectories, while allowing the facial and body dynamics to adapt naturally to the audio. Experiments show that the proposed method is able to produce plausible dubbing results.

**Strengths:**

* This paper goes beyond the traditional paradigm of mouth region editing, aiming to synchronize facial, head, and body movements with audio. The integration of key reference frames to preserve identity, gestures, and camera trajectory while enabling dynamic full-body motion is a promising advancement.

* The method is evaluated on multiple datasets (HDTF, CelebV-HQ, EMTD), using both quantitative and qualitative metrics. The model's ability to perform audio-aligned motion with high visual quality and identity preservation is validated through a series of objective and subjective tests.

**Weaknesses:**

* Figure 5 is confusing and difficult to understand. The caption of Figure 5 refers to "top 4 rows," but only 3 rows are visible in Figure 5 (left). Additionally, Figure 5 (right) lacks sufficient details for readers to understand how M0-M4 are implemented, and the meaning of the rectangles with different colors remains unclear. The figure should be corrected to match the caption, and a more detailed illustration about Sec.3.4 should be provided, clarifying the color-coding and the implementations of M0-M4.

* Key motions are missing after dubbing. The video in the supplementary material (supp/comparison/2_ours.mp4) demonstrates that key motions like "hands up" are missing after dubbing. This could results in loss of important content in the source videos, particularly in dynamic sequences.


* The source videos used for comparison are missing from the supplementary materials. This omission makes it difficult to judge how well the model maintains the original video’s motion, especially with respect to facial and body dynamics.

* The method described as sparse-frame dubbing appears to be a slightly modified version of the first-last-frame-to-video strategy, which limits its novelty. The paper could emphasize how sparse-frame dubbing significantly improves upon or differs from the first-last-frame strategy. A more detailed video comparison of the pros and cons of both approaches would be helpful.

* Figure 4 does not show how the future reference frame is placed during training and inference, leaving a gap in understanding how the model adapts across long sequences.

* The current supplemental materials are provided as separate videos, which makes it challenging to perform a side-by-side comparison. Additionally, the purpose of each video is unclear (e.g., I don't understand what ```comparison/0_0.mp4, 1_1.mp4, 2_2.mp4``` refers to) , and the input source videos are missing. I strongly recommend that the authors reorganize the supplemental videos into a single video, with clear captions to explain each segment. This will help highlight the key advantages of the proposed method and make the comparison more accessible.

**Questions:**

See [weaknesses]

---

### Note · Authors · 2025-11-14

I have read and agree with the venue's withdrawal policy on behalf of myself and my co-authors.